# Memory-efficient particle filter recurrent neural network for object localization

## Abstract

This study proposes a novel memory-efficient recurrent neural network (RNN) architecture specified to solve the object localization problem. This problem is to recover the object states along with its movement in a noisy environment. We take the idea of the classical particle filter and combine it with GRU RNN architecture. The key feature of the resulting memory-efficient particle filter RNN model (mePFRNN) is that it requires the same number of parameters to process environments of different sizes. Thus, the proposed mePFRNN architecture consumes less memory to store parameters compared to the previously proposed PFRNN model. To demonstrate the performance of our model, we test it on symmetric and noisy environments that are incredibly challenging for filtering algorithms. In our experiments, the mePFRNN model provides more precise localization than the considered competitors and requires fewer trained parameters.

## 1 Introduction

We consider the object localization problem and propose a novel GRU-like architecture to solve it. The object localization problem aims to filter the state vector of the object from the noise in motion and measurements. Typically standard GRU-like models (Shi & Yeung, 2018) are used to process sequential data, e.g. to predict the next item in a sequence, and classify or generate texts, audio, and video data. The considered object localization problem differs from the aforementioned problems since auxiliary data about the environment and particular measurements are available. Therefore, this additional knowledge should be incorporated into the GRU architecture properly to improve the model performance. Such a modification can be based on the existing approaches to solve the considered problem, which are discussed further.

One of the classical non-parametric methods to solve the object localization problem is particle filter (Gustafsson, 2010), which estimates the filtered object state from the states of auxiliary artificial objects that are called particles. A modification of GRU and LSTM recurrent neural networks with particle filter ingredients is presented in (Ma et al., 2020), where a particle filter recurrent neural network (PFRNN) is proposed. The core element of PFRNN is the modified cell (GRU or LSTM) equipped with analogs of particles and the corresponding weights of particles to estimate the filtered state. However, PFRNN does not consider specific features of the object localization problem and its performance can be improved with additional available data.

Therefore, we propose the novel *memory-efficient PFRNN (mePFRNN)* that combines the model assumptions used in the classical filtering methods (e.g. Kalman filter and particle filter) and parametrization from the GRU architecture. Such a combination provides more accurate state estimation and improves robustness in noisy and symmetric environments. Also, the Soft resampling procedure is used to avoid the degeneracy issue and improve the stability of the filtered states.

The main contributions of our study are the following.

1. We modified the general-purpose PFRNN model and proposed the mePFRNN model focused on the object localization problem.

2. The proposed mePFRNN model requires the same number of parameters for environments of different sizes.

3. We perform an extensive experimental comparison of our mePFRNN model with the existing RNN models and other non-parametric methods like the particle filter.

**Related works.** The object localization problem appears in a lot of applications like driving autonomous vehicles (Woo et al., 2018), navigation (Barczyk & Lynch, 2012; Lim et al., 2012), image processing (Costagli & Kuruoğlu, 2007), finance (Racicot & Théoret, 2010) and fatigue predictions (Yang et al., 2017). Therefore, there are a lot of different approaches to solving it. We can split them into two classes: non-parametric and parametric. The first class consists of classical methods that do not require a training stage and perform filtering of the object states on the fly. Examples of such methods are Kalman filter (Auger et al., 2013; Grewal & Andrews, 2010), and its modifications like extended (Julier & Uhlmann, 1997), unscented (Julier & Uhlmann, 2004), invariant extended (Bonnable et al., 2009) and ensembled (Houtekamer & Mitchell, 1998) Kalman filters. Also, methods related to the particle filter, e.g. multiparticle Kalman filter (Korkin et al., 2023), particle filters combined with genetic algorithms (Moghaddasi & Faraji, 2020), and particle swarm technique (Zhao & Li, 2010), box particle filter (Gning et al., 2013) and others are non-parametric filtering methods. The second class consists of parametric methods such that a pre-training stage is necessary before starting filtering. Such methods are typically based on neural networks that are trained on the collected historical data and then tested on the new data from real-world simulations. Although the pre-training stage may require a lot of time, one can expect that the inference stage, in which filtering is performed, is sufficiently fast due to modern hardware acceleration. Moreover, since the neural network models can efficiently treat sequential data (Wang et al., 2020a; LeCun et al., 2015), the parametric methods can provide more accurate filtering results compared to non-parametric methods.

Although the Transformer model (Vaswani et al., 2017) demonstrates superior performance over the considered GRU RNN in sequence processing tasks, it consumes a lot of memory to store parameters, requires special techniques for training (Gusak et al., 2022) and may not fit in the on-device memory limits. The memory-efficient Transformer models (Wang et al., 2020b; Kitaev et al., 2020; Jaszczur et al., 2021) may be a remedy for the observed issue and will be investigated in future work.

## 2 PROBLEM STATEMENT

Consider the trajectory of object states encoded as a sequence of $d$-dimensional vectors $\mathbf{x}_i \in \mathbb{R}^d$, where $i$ is an index of the time moment $t_i$. For example, if the object's state consists of 2D coordinates and 2D velocity, then state dimension $d = 4$. The states are changed according to the motion equation, which combines the physical law and the control system of the object. Formally we can write the motion equation as follows

$$\mathbf{x}_i = f(\mathbf{x}_{i-1}, \mathbf{u}_i, \boldsymbol{\eta}_i), \tag{1}$$

where $\mathbf{u}_i$ is a vector of control at the time moment $t_i$, for example, external forces, and $\boldsymbol{\eta}_i$ is a vector of noise corresponding to the object motion at the time moment $t_i$. Since the object moves with some noise, we should use additional measurements to estimate states more precisely. Typically there are several beacons in the environment, which are used by objects to measure some quantities that can improve their state estimate. For example, distance to the $k$-nearest beacons can improve the estimate of the object's location. Formally, denote by $\mathbf{y}_i \in \mathbb{R}^k$ a vector of measurements at time moment $t_i$ that is related with state estimate through the measurement function $g : \mathbb{R}^d \to \mathbb{R}^k$:

$$\mathbf{y}_i = g(\mathbf{x}_i, \boldsymbol{\zeta}_i), \tag{2}$$

where $\boldsymbol{\zeta}_i$ is the additional noise of measurement.

Object localization problem is the problem of estimating object trajectory from the given motion and measurement functions that represent the physical law of the environment and beacons' configuration, respectively. In this study, we introduce the parametric model $h_{\boldsymbol{\theta}} : \mathbb{R}^d \times \mathbb{R}^k \times \mathbb{R}^n \to \mathbb{R}^d$ that depends on the unknown parameters $\boldsymbol{\theta} \in \mathbb{R}^n$ and performs filtering of the inexact state estimate $\mathbf{x}_i$ based on the additional measurements $\mathbf{y}_i$. Assume we have training trajectory of the ground-truth states $\{\mathbf{x}_i^*\}_{i=1}^N$. Then we can state the optimization problem to fit our parametric model to the training data $\{\mathbf{x}_i^*\}_{i=1}^N$ and evaluate the generalization ability of the resulting model. In particular, the

standard loss function in such a problem is the mean square error loss function

$$MSE = \frac{1}{N} \sum_{i=1}^{N} \|h_{\boldsymbol{\theta}}(\mathbf{x}_i, \mathbf{y}_i) - \mathbf{x}_i^*\|_2^2 \tag{3}$$

such that the motion function $f$ and the measurement functions $g$ give the state estimate $\mathbf{x}_i$ and measurement vector $\mathbf{y}_i$, respectively.

We further focus on the plane motion setup, where the state vector consists of 2D coordinates $\mathbf{c} \in \mathbb{R}^2$ and a heading $\alpha \in [0, 2\pi]$, which defines the direction of movement, i.e. $\mathbf{x} = [\mathbf{c}, \alpha] \in \mathbb{R}^3$. Therefore, we follow (Ma et al., 2020) in slightly adjusting the MSE objective function equation 3 to treat coordinates and angles separately and compose the weighted MSE loss function:

$$wMSE = \underbrace{\frac{1}{N} \sum_{i=1}^{N} \|\mathbf{c}_i - \mathbf{c}_i^*\|_2^2}_{=\mathrm{MSE}_c} + \frac{\beta}{N} \sum_{i=1}^{N} (\alpha_i - \alpha_i^*)^2, \quad \text{where} \quad [\mathbf{c}_i, \alpha_i] = h_{\boldsymbol{\theta}}(\mathbf{x}_i, \mathbf{y}_i), \quad \mathbf{x}_i^* = [\mathbf{c}_i^*, \alpha_i^*],$$

$$\tag{4}$$

where $\beta > 0$ is a given weight. However, the $wMSE$ loss function treats angles $2\pi - \epsilon$ and $\epsilon$ as essentially different while they are physically close. Thus, we propose a novel modification of the mean squared loss function equation 3, that treats headings differently. In particular, we compare not angles but their sine and cosine in the following way:

$$L(\boldsymbol{\theta}) = \mathrm{MSE}_c + \frac{\beta}{N} \sum_{i=1}^{N} \left[ (\sin \alpha_i - \sin \alpha_i^*)^2 + (\cos \alpha - \cos \alpha_i^*)^2 \right], \tag{5}$$

where we use the same notation as in equation 4. Thus, we have the following optimization problem:

$$\boldsymbol{\theta}^* = \arg\min_{\boldsymbol{\theta}} L(\boldsymbol{\theta}),$$
$$\text{s.t. } \mathbf{x}_i = f(\mathbf{x}_{i-1}, \mathbf{u}_i, \boldsymbol{\eta}_i) \tag{6}$$
$$\mathbf{y}_i = g(\mathbf{x}_i, \boldsymbol{\zeta}_i).$$

Additionally to the MSE-like loss function, we evaluate the resulting model with the Final State Error (FSE) loss function, which reads as

$$\mathrm{FSE} = \|\mathbf{c}_N - \mathbf{c}_N^*\|_2, \tag{7}$$

where $[\mathbf{c}_N^*, \alpha_N^*] = \mathbf{x}_N^*$, $[\mathbf{c}_N, \alpha_N] = h_{\boldsymbol{\theta}}(\mathbf{x}_N, \mathbf{y}_N)$ and $t_N$ is a last-time moment in the considered period. Although the FSE loss function is widely used in previous studies (Ma et al., 2020; Zhu et al., 2020b), it may overestimate the filter performance due to the uncertainty in the filtering process. The final coordinates may be filtered very accurately by accident while filtering the previous coordinates may be quite poor. Thus, we focus on the $\mathrm{MSE}_c$ loss function as the main indicator of the filter performance.

The key ingredient of this approach is the selection of the proper parametric model $h_{\boldsymbol{\theta}}$. Following (Ma et al., 2020) we modify the GRU model such that it solves the object localization problem specifically. A detailed description of our modification is presented in the next section.

## 3 PARTICLE FILTER

One of the most efficient non-parametric approaches to solving the localization problem is the particle filter. This filter considers artificially generated particles with states $\mathbf{p}_i^{(k)} \in \mathbb{R}^d$ at the $i$-th time step and the corresponding weights $w_i^k \geq 0, \sum_{k=1}^{K} w_i^k = 1$ such that the estimate of the object state at the $i$-th time step is computed as follows

$$\hat{\mathbf{x}}_i = \sum_{k=1}^{K} w_i^k \mathbf{p}_i^{(k)},$$

where $K$ is the number of particles. Particles' weights are updated according to the corresponding measurements and state updates based on the Bayes rule and likelihood estimation, see (Chen et al.,

2003) for details. The important step in the particle filter is resampling, which corrects the updated particle weights and states to improve the accuracy of estimate $\hat{x}_i$. The resampling step addresses the degeneracy issue, which means a few number of particles have non-zero weights. This phenomenon indicates the poor representation of the target object state. The purely stochastic resampling samples particles' indices from the multinomial distribution according to the updated weights and then update particle states, respectively, see (8).

$$
\begin{aligned}
& i_1, \ldots, i_K \sim \text{Multinomial}(w_{i+1}^1, \ldots, w_{i+1}^K) \\
& \mathbf{p}_{i+1}^1, \ldots, \mathbf{p}_{i+1}^K \leftarrow \mathbf{p}_{i+1}^{i_1}, \ldots, \mathbf{p}_{i+1}^{i_K} \\
& w_{i+1}^k = \frac{1}{K}.
\end{aligned}
\tag{8}
$$

After resampling the resulting particle states are slightly perturbed with random noise to avoid equal particles' states. Since the particle filter processes sequential data through the recurrent updates of the particles and weights, the natural idea is to incorporate a similar approach in the recurrent neural network architecture. The particle filter recurrent neural network is proposed in (Ma et al., 2020) and we briefly describe it in the next section to highlight the difference with the proposed mePFRNN.

## 4 RECURRENT NEURAL NETWORKS INSPIRED BY PARTICLE FILTER

This section presents our RNN cell based on the particle filter idea, explicitly measured data, and beacons' positions. Since our model is a modification of the PFRNN (Ma et al., 2020) model, we briefly provide the main ingredients of this model.

**PFRNN.** Denote by $K$ a number of particles that are emulated in the PFRNN model. Below we consider motion $\mathbf{x}_i^{(k)}$ and measurement $\mathbf{y}_i^{(k)}$ vectors corresponding to the $k$-th particle at the $i$-th time moment, so $k = 1, \ldots, K$ and $i = 1, \ldots, N$. PFRNN considers the environment as a 2D array and constructs its embedding through the following encoder subnetwork:

$$
\text{Conv} \rightarrow \text{ReLU} \rightarrow \text{Conv} \rightarrow \text{ReLU} \rightarrow \text{Conv} \rightarrow \text{ReLU} \rightarrow \text{Flatten} \rightarrow \text{Linear} \rightarrow \text{ReLU}, \tag{9}
$$

where $\text{Conv}$ is a convolution layer, $\text{ReLU}$ denotes element-wise ReLU non-linearity, $\text{Linear}$ denotes a linear layer and $\text{Flatten}$ denotes a vectorization operation that reshapes the input tensor to a vector. The output of this subnetwork is the environment embedding vector $\mathbf{e}_{env}$. After that, the input to the PFRNN cell is constructed according to the following subnetwork:

$$
\begin{array}{c}
\mathbf{x}_i^{(k)} \longrightarrow \text{Linear} \rightarrow \text{ReLU} \rightarrow \odot \leftarrow \text{ReLU} \leftarrow \text{Linear} \leftarrow \text{ReLU} \leftarrow \text{Linear} \nwarrow \\
\downarrow \\
\mathbf{v}_i^{(k)} \longleftarrow \text{Concatenation} \qquad\qquad\qquad \mathbf{e}_{env} \\
\uparrow \\
\mathbf{y}_i^{(k)} \longrightarrow \text{Linear} \rightarrow \text{ReLU} \rightarrow \odot \leftarrow \text{ReLU} \leftarrow \text{Linear} \leftarrow \text{ReLU} \leftarrow \text{Linear} \swarrow
\end{array}
\tag{10}
$$

Note that, from (10) follows that the environment embedding $\mathbf{e}_{env}$ is explicitly used in the construction the input to the PFRNN cell. The baseline PFRNN cell is presented in both a graphical way (see Figure 1) and an analytical way (see equation (12)) for the reader's convenience. We note that this cell includes a reparametrization trick and updates not only the hidden states for every particle $\mathbf{h}_i^{(k)}$ but also the corresponding weights $w_i^{(k)}$ that are used in the resampling step. These weights typically correspond to the probability of the particle being equal to the ground-truth object state. However, in our experiment such weights are the logarithm of the corresponding probabilities, therefore the normalization step after update has the given form (see the last line in (12). After that, we adjust the resampling step to deal with the logarithms of the weights properly, see the paragraph below.

**Resampling procedure.** After the inference stage in the considered RNN cells, one has to make resampling, to mitigate the potential degeneracy problem. There are different approaches to performing resampling (Li et al., 2015; Zhu et al., 2020a). The main requirement for the resampling procedure in the parametric model is to be differentiable. Therefore, the stochastic resampling (8) is not directly fitted to the considered model. Instead, the Soft Resampling procedure (Ma et al., 2020) was proposed as a trade-off between the accuracy and the related costs. This approach to resampling considers a mixture of the distribution induced by weights and the uniform distribution with

probabilities $1/K$. Therefore, the formula for updating weights and hidden states reads as follows.

$$i_1, \ldots, i_K \sim \text{Multinomial}(\alpha w_{i+1}^1 + (1-\alpha)/K, \ldots, \alpha w_{i+1}^K + (1-\alpha)/K)$$

$$\mathbf{h}_{i+1}^1, \ldots, \mathbf{h}_{i+1}^K \leftarrow \mathbf{h}_{i+1}^{i_1}, \ldots, \mathbf{h}_{i+1}^{i_K} \tag{11}$$

$$w_{i+1}^k \leftarrow \frac{w_{i+1}^{i_k}}{\alpha w_{i+1}^{i_k} + (1-\alpha)/K},$$

where $\alpha > 0$ to make the operation differentiable. Note that similar to the stochastic resampling, the updated hidden states $\mathbf{h}_i^k$ are slightly perturbed. Section 5 provides more details on the usage of soft resampling in our experiments.

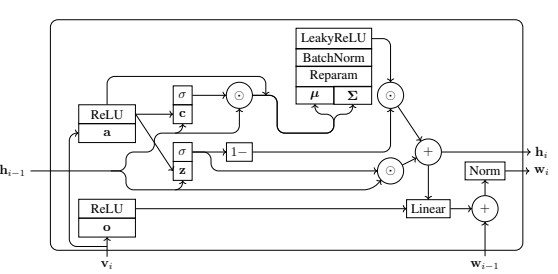

Figure 1: Baseline PFRNN cell design, where $\mathbf{a}, \mathbf{o}, \mathbf{z}, \mathbf{c}$ denote the linear layers that are used in computing the corresponding intermediate embeddings. For simplicity, we skip the superscript $k$ that indicates the particle index. This cell updates both particle hidden states and weights. Denote elementwise addition and multiplication by $+$ and $\odot$.

$$\mathbf{a}_i^{(k)} = \text{ReLU}(\boldsymbol{W}_a \mathbf{v}_i^{(k)} + \mathbf{b}_a)$$

$$\mathbf{o}_i^{(k)} = \text{ReLU}(\boldsymbol{W}_o \mathbf{v}_i^{(k)} + \mathbf{b}_o)$$

$$\mathbf{c}_i^{(k)} = \sigma(\boldsymbol{W}_c[\mathbf{a}_i^{(k)}, \mathbf{h}_{i-1}^{(k)}] + \mathbf{b}_c) \odot \mathbf{h}_{i-1}^{(k)}$$

$$\mathbf{z}_i^{(k)} = \sigma(\boldsymbol{W}_z[\mathbf{a}_i^{(k)}, \mathbf{h}_{i-1}^{(k)}] + \mathbf{b}_z)$$

$$\boldsymbol{\mu}_i^{(k)} = \boldsymbol{W}_{\boldsymbol{\mu}}[\mathbf{a}_i^{(k)}, \mathbf{c}_i^{(k)}] + \mathbf{b}_{\boldsymbol{\mu}}$$

$$\boldsymbol{\Sigma}_i^{(k)} = \boldsymbol{W}_{\boldsymbol{\Sigma}}[\mathbf{a}_i^{(k)}, \mathbf{c}_i^{(k)}] + \mathbf{b}_{\boldsymbol{\Sigma}}$$

$$\epsilon \sim \mathcal{N}(0, \boldsymbol{I})$$

$$\mathbf{d}_i^{(k)} = \text{LeakyReLU}(\text{BN}(\boldsymbol{\mu}_i^{(k)} + \boldsymbol{\Sigma}_i^{(k)} \odot \epsilon))$$

$$\mathbf{h}_i^{(k)} = (1 - \mathbf{z}_i^{(k)}) \odot \mathbf{d}_i^{(k)} + \mathbf{z}_i^{(k)} \odot \mathbf{h}_{i-1}^{(k)}$$

$$\mathbf{p}_i^{(k)} = \boldsymbol{W}_w[\mathbf{o}_i, \mathbf{h}_i^{(k)}] + b_w$$

$$w_i^{(k)} = \mathbf{p}_i^{(k)} + w_{i-1}^{(k)} -$$
$$- LogSumExp(\mathbf{p}_i^{(k)} + w_{i-1}^{(k)}) \tag{12}$$

**mePFRNN.** Since PFRNN encodes the environment with the convolution operation, it requires training a number of parameters proportional to the environment size. To reduce the number of trainable parameters, we do not use the data about an environment as input to our model. Thus, the subnetwork (9) is removed from the mePFRNN architecture. We expect that beacons' and obstacles' positions, have to be implicitly extracted in the training stage of mePFRNN since the environment is the external factor to the localization problem and stays the same over the particular trajectory. The motion and measurement vectors corresponding to every particle are embedded into a high dimensional space via linear layers and ReLU non-linearity. Then, the obtained embeddings are concatenated and processed by a linear layer with LeakyReLU non-linearity. The result of the latter operation is motion embedding $\mathbf{e}_{\mathbf{u}}^{(k)}$ for every particle, which is additional input to the proposed mePFRNN cell. The encoding procedure described above is summarized in scheme (13), which is alternative to the scheme (10) in the baseline.

$$\mathbf{x}_i^{(k)} \to \text{Linear} \to \text{ReLU} \searrow$$
$$\quad\quad\quad\quad \text{Concatenation} \to \text{Linear} \to \text{LeakyReLU} \to \mathbf{e}_{\mathbf{u}}^{(k)} \tag{13}$$
$$\mathbf{y}_i^{(k)} \to \text{Linear} \to \text{ReLU} \nearrow$$

Thus, mePFRNN is a voxel-independent model that can be easily used in very large environments without increasing the number of trainable parameters. One more benefit of the proposed approach becomes crucial if the beacons in the environment are located not in the middle of the artificially generated voxels in the PFRNN model. These voxels compose a grid for the considered environment to identify the beacons and obstacles with convolution encoding. In this case, the convolution operation does not adequately encode the beacons' positions and makes further filtering more noisy. The resulting cell is shown in Figure 2 graphically and in equations (14) analytically, where $MLP$ consists of two sequential linear layers and intermediate LeakyReLU nonlinearity. Note that, the Soft Resampling procedure is also used here similar to the PFRNN model described above. Thus, the key differences between the baseline PFRNN and mePFRNN are summarized below:

- mePFRNN does not include the subnetwork (9) that constructs embedding of the entire environment;

- The input to the mePFRNN cell is constructed with the subnetwork (13) instead of the more complicated subnetwork (10);

Such modifications lead to the mePFRNN model that does not depend on the environment size and therefore is memory-efficient with respect to the increasing environment size.

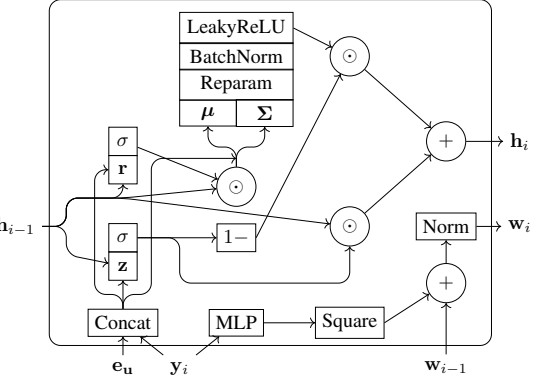

$$\mathbf{e}_i^{(k)} = [\mathbf{e_u}^{(k)}, \mathbf{y}_i^{(k)}]$$
$$\mathbf{z}_i^{(k)} = \sigma(\boldsymbol{W}_z[\mathbf{h}_{i-1}^{(k)}, \mathbf{e}_i^{(k)}] + \mathbf{b}_z)$$
$$\mathbf{r}_i^{(k)} = \sigma(\boldsymbol{W}_r[\mathbf{h}_{i-1}^{(k)}, \mathbf{e}_i^{(k)}] + \mathbf{b}_r)$$
$$\boldsymbol{\mu}_i^{(k)} = \boldsymbol{W}_{\boldsymbol{\mu}}[\mathbf{r}_i^{(k)} \odot \mathbf{h}_{i-1}^{(k)}, \mathbf{e}_i^{(k)}] + \mathbf{b}_{\boldsymbol{\mu}}$$
$$\boldsymbol{\Sigma}_i^{(k)} = \boldsymbol{W}_{\boldsymbol{\Sigma}}[\mathbf{r}_i^{(k)} \odot \mathbf{h}_{i-1}^{(k)}, \mathbf{e}_i^{(k)}] + \mathbf{b}_{\boldsymbol{\Sigma}}$$
$$\epsilon \sim \mathcal{N}(0, \boldsymbol{I})$$
$$\mathbf{d}_i^{(k)} = \text{LeakyReLU}(\text{BN}(\boldsymbol{\mu}_i^{(k)} + \boldsymbol{\Sigma}_i^{(k)} \odot \epsilon))$$
$$\mathbf{h}_i^{(k)} = (1 - \mathbf{z}_i^{(k)}) \odot \mathbf{d}_i^{(k)} + \mathbf{z}_i^{(k)} \odot \mathbf{h}_{i-1}^{(k)}$$

Figure 2: The proposed mePFRNN cell. Square block means elementwise square of the input. MLP consists of two linear layers and LeakyReLU intermediate non-linearity. Other notation is similar to Figure 1.

$$w_i = MLP(\mathbf{y}_i^{(k)})^2 + w_{i-1} -$$
$$- LogSumExp(MLP(\mathbf{y}_i^{(k)})^2 + w_{i-1})$$
$$(14)$$

**Alternative GRU-based models.** In addition to the proposed mePFRNN model, we also propose two approaches to exploiting the classical GRU model (see Figure 3 and equations (15)) in the object localization problem. Namely, the EnsembleGRU model consists of many small GRU cells whose predictions are averaged to estimate the target object state. The number of models in the ensemble and the number of trained parameters in every model are selected such that the total number of the trained parameters is approximately equal to # parameters in PFRNN times # particles. The complementary approach is just to use the single GRU cell, where the number of trained parameters is equal to # particles times # parameters in PFRNN. Both approaches are complementary to the PFRNN and mePFRNN models since they do not exploit particles. Also, note that the input to the GRU cell in EnsembleGRU and HeavyGRU models is the same as the input to the PFRNN cell.

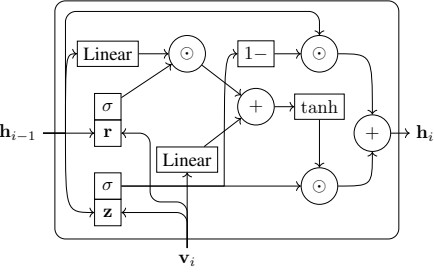

$$\mathbf{z}_i = \sigma(\boldsymbol{W}_z[\mathbf{v}_i, \mathbf{h}_{i-1}] + \mathbf{b}_z)$$
$$\mathbf{r}_i = \sigma(\boldsymbol{W}_r[\mathbf{v}_i, \mathbf{h}_{i-1}] + \mathbf{b}_r)$$
$$\hat{\mathbf{h}}_i = \tanh(\boldsymbol{W}_h\mathbf{v}_i + \mathbf{b}_h + \qquad (15)$$
$$\mathbf{r}_i \odot (\boldsymbol{U}_h\mathbf{h}_{i-1} + \mathbf{b}_u))$$
$$\mathbf{h}_i = (1 - \mathbf{z}_i) \odot \mathbf{h}_{i-1} + \mathbf{z}_i \odot \hat{\mathbf{h}}_i$$

Figure 3: Standard GRU cell, where $\mathbf{z}$ and $\mathbf{r}$ denote the linear layers to compute $\mathbf{z}_i$ and $\mathbf{r}_i$, Linear denotes linear layers to compute $\hat{\mathbf{h}}_i$, see (15).

## 5 COMPUTATIONAL EXPERIMENT

In this section, we demonstrate the performance of our model and compare it with alternative neural networks and non-parametric models. For training the compared neural networks we use RMSProp optimizer (Tieleman & Hinton, 2012) since it shows more stable convergence compared to Adam (Kingma & Ba, 2014) and SGD with momentum (Goodfellow et al., 2016), learning rate

equal to $5 \cdot 10^{-4}$ and every batch consists of 150 trajectories. The maximum number of epochs is 5000 for the considered environments. During the training stage, a validation set of trajectories is used to identify the overfitting. Therefore, different environments require a different number of epochs before overfitting occurs. In particular, overfitting does not occur after 5000 epochs in the world $10 \times 10$. At the same time, overfitting is observed after 600 and 200 epochs in the World $18 \times 18$ and WORLD $27 \times 27$, respectively.

**Trajectories generation procedure.** To evaluate the considered methods and demonstrate the performance of the proposed mePFRNN, we consider four environments, see Figure 4. Environments world $10 \times 10$, World $18 \times 18$, and WORLD $27 \times 27$ are symmetric and therefore challenging for object localization since symmetric parts can be confused by a filtering method. Environment *Labyrinth* is not symmetric and medium challenging for filtering methods. Thus, the considered filtering methods are compared comprehensively due to the diversity in the testing environments.

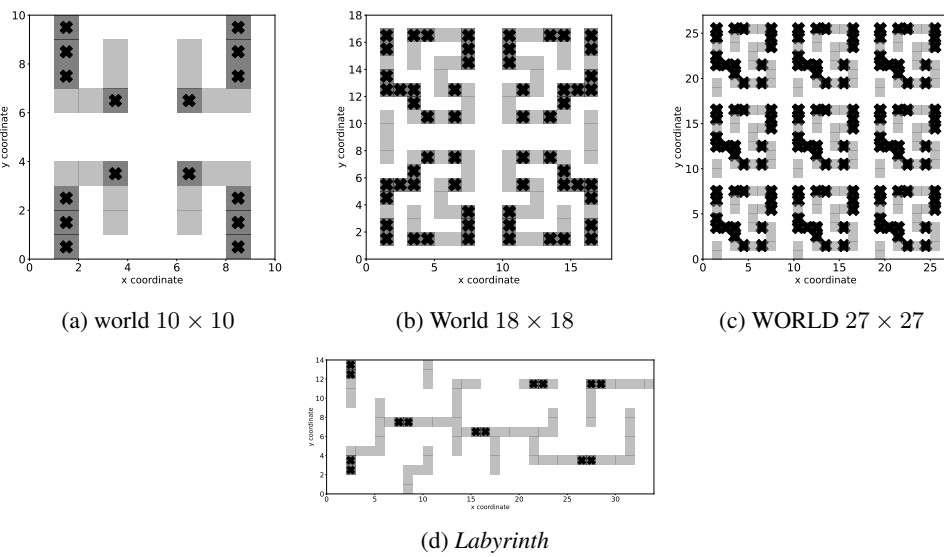

(a) world $10 \times 10$          (b) World $18 \times 18$          (c) WORLD $27 \times 27$

(d) *Labyrinth*

Figure 4: Visualization of test environments. Black crosses denote beacons, and grey blocks denote obstacles. The upper row represents the very symmetric environments that are especially challenging for solving the localization problem. The *Labyrinth* environment is not symmetric and is similar to the environment, which was used for the evaluation of filtering methods in (Zhu et al., 2020b).

To train the parametric models we need to generate a set of trajectories $\{\mathbf{x}_i^*\}_{i=1}^N$. Since our tests assume that the object's initial state is unknown, we set the initial state $\mathbf{x}_0^*$ randomly for all generated trajectories. Initial states do not intersect with obstacles. Then, every next iteration updates the object state according to the motion equation, where external velocity $u \in [0, 0.2]$ is known and the direction is preserved from the previous step within the noise. In the case of a collision with an obstacle, the object's direction is changed randomly such that the next state does not indicate the collision. To simulate engine noise, the velocity $u$ is perturbed by $\eta_r \sim \mathcal{U}[-0.02, 0.02]$. To simulate uncertainty in the object control system, the direction $\phi$ is also perturbed by $\eta_\phi \sim 2\pi\alpha$, where $\alpha \sim \mathcal{U}[-0.01, 0.01]$. The measurements $\mathbf{y}_i$ are the distances to the five nearest beacons, which are also noisy with the noise distributed as $\zeta \sim \mathcal{U}[-0.1, 0.1]$. In the considered environments, we set the number of time steps in every trajectory $N = 100$.

To train the considered parametric models, we generate 8000 trajectories, 1000 trajectories for validation, and an additional 10000 trajectories for the testing stage. During the training process, the MSE loss is computed for the validation trajectories and if the obtained value is smaller than the current best one, then the best model is updated. This scheme helps to store the best model during the training and avoid overfitting.

**The list of compared models.** We compare the proposed mePFRNN model with the following competitors combined in two groups. The first group consists of alternative recurrent neural net-

works that can solve the object localization problem, in particular the baseline PFRNN model from (Ma et al., 2020), HeavyGRU, and EnsembleGRU models. Following the study (Ma et al., 2020) we use the $wMSE$ loss function equation 4 to train alternative neural network models and use $L$ loss function equation 5 to train the proposed mePFRNN model. Such a choice of training setup highlights the benefit of the proposed loss function $L$. In both settings, we use $\beta = 0.1$.

The second group consists of the particle filter (PF) and the multiparticle Kalman filter (MKF). We include these methods in the experiments to compare the performance of the parametric and non-parametric models. The performance is measured in terms of $MSE_c$, FSE, number of trained parameters, training time, and inference time. Note that, non-parametric models do not require training, therefore they are more lightweight. However, to get high accuracy a lot of particles are needed which leads to long runtime. Thus, for adequate comparison with neural methods, the classical filters were used with fewer particles to show a similar runtime as neural network-based models in the inference mode. In addition, we use stochastic resampling in the non-parametric models and Soft Resampling in the parametric ones. However, the Soft Resampling procedure for the non-parametric models does not significantly change the final performance. The comparison of the aforementioned models is presented in the next paragraph.

**Discussion of the results.** In experiment evaluation, we compare non-parametric and parametric models with the four test environments described above. The obtained results are summarised in Table 1. Also, we track the number of trained parameters, the amount of memory that is necessary to store them, and the runtime to update the object state in one step. From this table follows that the proposed mePFRNN model gives the best or the second-best $MSE_c$ score for the considered environments. At the same time, the FSE score is typically smaller for HeavyGRU or EnsembleGRU in the considered environments. One more important factor is the number of trainable parameters. The smaller the number of parameters, the easier embedding the model in hardware. The mePFRNN model requires fewer trainable parameters compared with other parametric models, i.e. PFRNN, HeavyGRU, and EnsembleGRU. The last but not least feature of the considered models is the inference time, i.e. the runtime to update the object state from the $i$-th to the $(i + 1)$-th time step. mePFRNN is slightly slower than PFRNN, and HeavyGRU appears the fastest model in the inference stage. Thus, we can conclude that the proposed mePFRNN model provides a reasonable trade-off between $MSE_c$ score, number of trainable parameters, and inference time among the considered parametric and non-parametric models tested in the selected benchmark environments.

The number of particles chosen in Table 2 is such that the inference runtime is close to the inference runtime of the considered neural networks. Since in Table 1 we fix the particular number of particles in non-parametric models, we present the $MSE_c$ and FSE losses for the larger number of particles in Table 2. It shows that if the number of particles is sufficiently large, both $MSE_c$ and FSE values are smaller than the corresponding values for parametric models. However, such an accurate estimation of states requires a much slower inference runtime compared to the considered parametric models. Thus, the neural network-based filters are of significant interest since they can show better accuracy compared to non-parametric models and provide faster updates of the object's state.

## 6 CONCLUSION

We present the novel recurrent neural network architecture mePFRNN to solve the object localization problem. It combines the standard GRU RNN, particle filter, and explicit measurements of distances from the object to the beacons. The latter feature makes the proposed model memory-efficient since the number of trainable parameters does not depend on the environment size. We compare the proposed mePFRNN model with the general-purpose PFRNN model and two modifications of standard GRU RNN. The test environments consist of symmetric environments of different sizes and the non-symmetric *Labyrinth* environment. Such diversity of the test environments leads to the comprehensive comparison of the considered parametric models to solve the object localization problem. Although mePFRNN appears slightly slower in inference than the baseline PFRNN, it filters the object's coordinates more precisely in the considered symmetric environments. Moreover, mePFRNN does not exploit explicit data about the environment or the corresponding embeddings. At the same time, the proposed mePFRNN model outperforms competitors in MSE values for the most of considered test environments.

Table 1: Performance comparison of the filtering methods. Mean values averaged over 10000 runs and standard deviations in braces. The number of particles for PF and MKF is selected such that their filtering time is close to the inference time in neural network models. Dashes indicate non-parametric models, which do not have any trainable parameters and therefore do not consume memory. Note that we report mean $MSE_c$ and FSE values corresponding to object position only and ignore the angle component of the state vector. To estimate statistical significance between results of mePFRNN and competitors, we use t-test and obtain $p$-values that are less or equal $10^{-5}$. Therefore, decreasing of $MSE_c$ observed almost in all considered environments is significant.

| Environment | Model | $MSE_c$ | FSE | # parameters | # particles | Memory, Mb | Inference time, ms. |
|---|---|---|---|---|---|---|---|
| world $10 \times 10$ | mePFRNN (our) | **0.77** (**3.97**) | 0.13 (0.09) | 28802 | 30 | 0.46 | 1.0 |
| | PFRNN | 1.30 (5.70) | 0.05 (0.18) | 99472 | 30 | 1.5 | 1.1 |
| | HeavyGRU | 1.01 (5.31) | 0.07 (0.13) | 2453239 | 1 | 37 | 0.4 |
| | EnsembleGRU | 1.24 (4.49) | **0.04**(**0.13**) | 95283 | 30 | 45 | 4.4 |
| | PF | 13.70 (25.47) | 1.73 (2.81) | – | 200 | – | 2.0 |
| | MKF | 10.77 (23.87) | 1.37 (2.60) | – | 50 | – | 4.7 |
| World $18 \times 18$ | mePFRNN (our) | 7.08 (25.69) | 0.51 (0.63) | 28802 | 30 | 0.46 | 1.0 |
| | PFRNN | 10.74 (29.57) | 0.30 (0.71) | 214160 | 30 | 1.6 | 1.1 |
| | HeavyGRU | **6.83** (**26.13**) | **0.22**(**0.49**) | 2682615 | 1 | 41 | 0.4 |
| | EnsemleGRU | 9.79 (22.04) | 0.24 (0.54) | 209971 | 30 | 99 | 4.3 |
| | PF | 74.17 (91.80) | 5.73 (5.37) | – | 200 | – | 2.4 |
| | MKF | 96.13 (114.14) | 7.08 (6.57) | – | 50 | – | 4.6 |
| WORLD $27 \times 27$ | mePFRNN (our) | **59.65** (**55.59**) | **5.52** (**3.76**) | 28802 | 30 | 0.46 | 1.0 |
| | PFRNN | 68.28 (64.16) | 5.86 (3.96) | 465392 | 30 | 7.1 | 1.1 |
| | HeavyGRU | 73.36 (67.87) | 6.22 (3.99) | 3169367 | 1 | 48 | 0.4 |
| | EnsembleGRU | 67.41 (59.13) | 5.86 (3.67) | 461203 | 30 | 220 | 4.3 |
| | PF | 181.75 (171.81) | 11.09 (6.76) | – | 200 | – | 2.8 |
| | MKF | 200.36 (201.07) | 12.02 (7.60) | – | 50 | – | 6.8 |
| *Labyrinth* | mePFRNN (our) | **1.43** (**13.27**) | 0.30 (0.24) | 28802 | 30 | 0.46 | 1.0 |
| | PFRNN | 6.26 (29.28) | 0.18 (0.11) | 307696 | 30 | 2.5 | 1.1 |
| | HeavyGRU | 1.78 (13.80) | **0.12** (**0.11**) | 2838263 | 1 | 45 | 0.4 |
| | EnsembleGRU | 5.57 (5.66) | 0.12 (0.08) | 303507 | 30 | 135 | 4.4 |
| | PF | 87.23 (163.00) | 4.74 (6.92) | – | 200 | – | 1.8 |
| | MKF | 77.90 (169.20) | 4.19 (7.40) | – | 50 | – | 4.4 |

Table 2: Dependence of the PF and MKF performance and inference time to update the state vector on the number of particles. The more particles are used in these filters, the more accurate trajectories are recovered and the slower filtering is. Here we focus only on the loss functions that evaluate the accuracy of object coordinates filtering.

| Environment | Filter | # particles | $MSE_c$ | FSE | Inference time, ms. |
|---|---|---|---|---|---|
| world $10 \times 10$ | PF | 200 | 13.70 (25.47) | 1.73 (2.81) | 0.8 |
| | MKF | 50 | 10.77 (23.87) | 1.37 (2.60) | 4.7 |
| | PF | 10000 | 0.58 (3.67) | 0.20 (0.09) | 45 |
| | MKF | 10000 | 0.82 (3.75) | 0.20 (0.03) | 58 |
| World $18 \times 18$ | PF | 200 | 74.17 (91.80) | 5.73 (5.37) | 2.4 |
| | MKF | 50 | 96.13 (114.14) | 7.08 (6.57) | 4.6 |
| | PF | 10000 | 3.04 (18.24) | 0.22 (0.51) | 48 |
| | MKF | 10000 | 3.25 (13.89) | 0.21 (0.11) | 66 |
| WORLD $27 \times 27$ | PF | 200 | 181.75 (171.81) | 11.09 (6.76) | 2.8 |
| | MKF | 50 | 200.36 (201.07) | 12.02 (7.60) | 6.8 |
| | PF | 10000 | 74.86 (79.48) | 5.94 (5.06) | 37 |
| | MKF | 10000 | 55.83 (52.68) | 5.16 (3.93) | 50 |
| *Labyrinth* | PF | 200 | 87.23 (163.00) | 4.74 (6.92) | 1.8 |
| | MKF | 50 | 77.90 (169.20) | 4.19 (7.40) | 4.4 |
| | PF | 10000 | 1.53 (15.06) | 0.50 (0.02) | 40 |
| | MKF | 10000 | 1.50 (14.81) | 0.50 (0.03) | 90 |

## REPRODUCIBILITY

To reproduce the presented results, we attach the source code in zip-archive format with README file inside. This file consists of comprehensive instructions on running code and corresponding

hyperparameters for both considered models and optimizers. Upon acceptance, the source code will be released in the public GitHub repository.

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
