# OpenReview forum: "Memory-efficient particle filter recurrent neural network for object localization"
_ICLR.cc/2024/Conference — Submitted to ICLR 2024_

### Official Review · Reviewer_Y3Q6 · 2023-10-31

**Soundness:** 2 fair
**Presentation:** 2 fair
**Contribution:** 2 fair
**Rating:** 3
**Confidence:** 3

**Summary:**

This paper seems to have proposed some detailed modifications to the PFRNN framework (particle-filter based RNN).

**Strengths:**

The performance seems to be marginally better than PFRNN.

**Weaknesses:**

From what I gathered, the solution the authors provided is to reduce the number of parameters from PFRNN by avoiding the convolutional encoding of the environment factors. The authors argued about that PFRNN’s parameters are proportional to the environment size, however this is true only because PFRNN has a flatten layer which flattens the environment into a long vector. Such a “flatten” approach greatly inflates the number of parameters and have been abandoned in the deep learning community since the inception of ResNet in 2015. Hence, I think this approach ought to make a baseline comparison to the global average pooling approach that has been adopted since ResNet, if not newer transformer-based approaches. If one uses global average pooling instead of flatten, then PFRNN’s # of parameters won’t scale with environment size, and we would have a much more fair comparison between the baseline and the proposed approach.

There are some confusing parts in the comparison to PFRNN too. In eq. (12), the authors presented their approach of encoding the x and y as being novel, however I don’t see a significant difference between that and the paragraph after eq. (9), where we also see PFRNN encoding x and y after 2 linear layers as n and m. Is there a significant difference between this and eq. (12) besides the fact that concatenation is done at a slightly different place?

Besides, there are some other differences such as multiplying some state feature vectors instead of concatenating them, which I don’t think was discussed in enough detail about the motivation and the performance gains it provides.

Also, I am not convinced, from the standard deviation numbers the authors provided, that this approach is truly better than the baselines. The standard deviations are routinely 3-4 times of the mean, which I believe would lead to any statistical tests to indicate most of the differences shown in the tables to be statistically insignificant.

Other issues:

I don’t agree calling eq. (8) “perturbation with random noise” that seems to be misleading. What eq. (8) really does is just a resampling of the particles with replacement. It would be better to clarify the terminology.

The font size of Fig. 1 and Fig. 2 are too small to be legible. Besides, the presentation approach of showing huge sets of equations with changes of variable names (between e.g. eq. (11) and eq. (13) and just hoping that the readers will figure out what is different is not a great practice. It would be much easier if the authors delineate clearly what are the differences between eq. (11) and eq. (13).

Overall:

In general, the authors have not shown sufficient evidence to convince me that this algorithm they proposed is novel enough, or it significantly reduces the number of parameters and the model size, or whether it improves the results or not. I also do not believe this work is well-presented, because of its resolution on having readers to read through long equations. Hence, I cannot argue for this paper’s acceptance at this stage.

**Questions:**

Please compare with baselines that do not use the flatten approach in the final layer, such as any ResNet-based algorithms or newer transformer-based approaches that employ global average pooling.

Please clarify the difference between eq. (12) and the paragraph after eq. (9).

Please discuss the motivation of the design choices of multiplying state vectors.

Please discuss the variance in the results.

---

> ### Author Response · Authors · 2023-11-23
>
> Dear reviewer Y3Q6,
>
> thanks for the detailed comments and careful reading of our manuscript! Below we present our responses to your comments and modify the main text respectively.
>
> 1) We agree that the Flatten layer leads to a large number of trainable parameters. The motivation to use it in the baseline PFRNN is unclear to us, the authors do not provide a detailed explanation in the original paper [Ma et al, 2020]. At the same time, we kindly disagree that incorporating average pooling in the baseline PFRNN architecture is a remedy to the observed issues. We perform comparison tests on how the number of trainable parameters depends on the environment size for the PFRNN equipped with average pooling layers in between convolutional layers as it is done in ResNet models. The results are presented in the Table below.
> | The size of the square environments, which are similar to the one presented in the manuscript | Number of trainable parameters in PFRNN with intermediate average pooling layers |
> |:------:|:------:|
> | $50 \times 50$ | 24568 |
> | $100 \times 100$ | 61560 |
> | $200 \times 200$ | 172408 |
> | $400 \times 400$ | 320120 |
> | $800 \times 800$ | 467832 |
> |  $1600 \times 1600$ | 615544 |
>
>
> So, the number of trainable parameters grows approximately as the square root of the environment size.
> Thus, instead of reducing the observed dependence, we suggest the mePFRNN model that completely does not depend on the environment size.
> In contrast to the PFRNN approach (even with average pooling), we do not construct environment embedding and process only motion and measurement vectors. Thus, our mePFRNN model by construction requires less number of trainable parameters and demonstrates superior performance in the benchmark environments.
>
> 2) To highlight the difference between the processing of motion and measurement vectors in the baseline PFRN and our mePFRNN, we provide scheme (10) corresponding to the baseline PFRNN. We replace this scheme with scheme (13) in our model, where environment embedding is not used. Also, we do not compute elementwise products between embeddings only concatenate them. Also, we summarize the key differences between mePFRNN and PFRNN at the end of the mePFRNN paragraph slot.
>
> 3) We provide standard deviations for the obtained MSE_c only for comparison purposes. In particular, one can observe that our model (mePFRNN) gives less standard deviation than baseline PFRNN. Although the standard deviation is 3-4 times larger than the reported mean MSE_c, it does not indicate that the difference between them is statistically insignificant. Since we have generated 10000 trajectories with 100 points in every trajectory, we run the two-sided t-test to estimate the significance of the mean MSE_c difference and obtain p-values less than 10^{-5}. This value indicates that the difference between mean MSE_c is statistically significant. We revised the caption to Table 1, respectively.
> In addition, note that the reported standard deviation is directly connected with the size of the environment. In particular, in the environment WORLD 27x27 standard deviation between two random points is approximately equal to 200, which is much larger than the reported standard deviation $\approx$ 50.
>
> 4) We are sorry for the confusing description of eq (8). It is indeed about only resampling stage. After this step, states are perturbed with random noise. We have fixed the main text and mentioned the random perturbation after eq (8).
>
> 5) We have added the list of key differences between the mePFRNN model and the baseline PFRNN explicitly. Also, we have updated the font size in the cell pictures. These pictures and corresponding equations are presented not for direct comparison but for completeness of the explanation.
>
> 6) The multiplications of embedding vectors appear only in the baseline PFRNN and the authors do not provide clarifications in the corresponding paper [Ma, et al, 2020]. We assume that this operation aims to combine the embedding vector for the environment with state and motion embeddings such that the resulting state estimates take into account the environment structure. In our approach, we exclude the environment embedding and just concatenate embeddings for motion and measurement and process them jointly.
>
> Thus, we kindly ask you to take into account the submitted response with the revision of the manuscript and increase the score for our study.

---

### Official Review · Reviewer_B8nM · 2023-11-01

**Soundness:** 2 fair
**Presentation:** 1 poor
**Contribution:** 1 poor
**Rating:** 1
**Confidence:** 4

**Summary:**

This paper introduces a memory-efficient Particle Filter RNN architecture for object localisation. The proposed method builds upon the existing PFRNN work (which itself introduces a modified RNN/GRU cell equipped with particle states and weights), incorporating additional assumptions from classical estimation methods (KFs and PFs).

**Strengths:**

- Good theoretical development.
- Interesting research direction to blend ideas from classical methods into learning architectures.

**Weaknesses:**

- Unfortunately, there doesn’t appear to be much technical novelty of the proposed mePFRNN compared with PFRNN. The main difference is that PFRNN encodes the entire environment as a 2D grid, while mePFRNN does not require this, and operates on motions and states directly. I feel this could be a simple model variant rather than a novel technical contribution that warrants a separate publication.
- The method is evaluated in quite simple grid world environments, in a narrow setting with beacons placed in the environment. The method would be more convincing if applied to larger-scale data or without specific observations relating to beacons or obstacles.

**Questions:**

- I found the paper quite hard to follow in places. For the introduction, a more linear narrative through each paragraph would be helpful. Eg. first paragraph on the problem to be solved, the second one on existing methods, and third one on the proposed approach leading into the contributions.
- A lot of the citations need to be fixed to use \citep{}; many of them blend into the rest of the text without parentheses.

---

> ### Author Response · Authors · 2023-11-23
>
> Dear reviewer B8nM,
>
> Thanks for your comments and suggestions on improving the quality of the manuscript! Below we provide responses to your remarks and questions and include them in the main text revision (see updated pdf).
> 1) We kindly disagree that the modification of PFRNN is technical since the proper specification of the general-purpose PFRNN to the object localization problem is not obvious. The processing data from beacons and positions is not straightforward and requires additional tests that are presented in our study. Moreover, we perform an extensive comparison of GRU-based RNNs with classical filtering methods to demonstrate the trade-off between model size and resulting performance.
> 2) We selected the test environments such that they are challenging for the classical non-parametric particle filter method. Although the considered environments are grid worlds, they are not simple for filtering methods since in the case of the unknown initial state, a filtering method may confuse the symmetric blocks of the grid and fail to recover the correct state of the object.
> Our study considers a particular filtering problem stated formally in eq (6). We agree that other problem statements related to the filtering problem are possible, however, in this study, we focus on the particular form of the filtering problem and compare the methods in the corresponding setup.
> 3) We updated the citation style and revised the introduction section according to the above suggestions.
>
> Thus, we kindly ask you to take into account the aforementioned response and increase the score.

---

### Official Review · Reviewer_mq82 · 2023-11-08

**Soundness:** 3 good
**Presentation:** 2 fair
**Contribution:** 3 good
**Rating:** 8
**Confidence:** 3

**Summary:**

The study introduces an innovative approach to object localization in noisy environments through a new recurrent neural network (RNN) design. By integrating the classical particle filter with the gated recurrent unit (GRU) RNN framework, the researchers have developed a memory-efficient particle filter RNN model (mePFRNN). Notably, the mePFRNN boasts an invariant parameter count regardless of the environment size, marking a substantial improvement in memory efficiency over its predecessor, the PFRNN model. The model's effectiveness was evaluated in symmetrical and noisy conditions, where traditional filtering algorithms typically struggle. The findings indicate that the mePFRNN outperforms existing models in accuracy while also requiring fewer parameters to be trained, highlighting its potential for more efficient and precise object localization tasks.

**Strengths:**

- The paper presents an effective method to tackle the memory bottleneck for large scale object localization problems.
- Using neural networks to encode the spatial information so that the representation becomes voxel-independent is promising and novel in the context of particle filtering.
- The paper is well written and well formatted.
- The authors demonstrate certain advantages on reduced memory. It would be useful for robotics that run inference on real time with limited memory constraints.
- Proper ablation studies have been done and discussed.

**Weaknesses:**

- The presentation could be improved by providing a better comparison picture side-by-side between the proposed method and the PFRNN method.
- [Minor] Table1 and 2 should add optimal direction (up or down) in each column of the table.

**Questions:**

Consider the case of real world robot localization. How large would the memory requirement to be? Table1 suggests that the memory requirement for PFRNN is not very much with less number of particles. How much gain would the proposed method gain when scaling up to a larger problem comparing to PFRNN?

---

> ### Author Response · Authors · 2023-11-23
>
> Dear reviewer mq82,
>
> Thanks for your comments on our study! Below we provide a point-to-point response.
> 1) In our work, we show that the number of trainable parameters in the mePFRNN model does not depend on the environment size in contrast to the previously proposed PFRNN model. We demonstrate this in challenging symmetric environments, where we can control the number of beacons, their locations, and also the map of the obstacles. Regarding real-world robot localization, we consider as one of the main directions for future work since currently it is hard to find the proper simulator that generates such environments.
>
> 2) The key feature of the proposed mePFRNN model is that its number of parameters does not depend on the environment size in contrast to the baseline PFRNN model. Thus, the larger problems can be solved with the mePFRNN model, which has the same number of trainable parameters.
>
> 3) We have revised the text to improve readability and highlight the difference between the proposed mePFRNN model with the baseline PFRNN.

---

### Meta-Review · Area_Chair_BVLW · 2023-12-09

**Metareview:**

The paper proposes a variation of a Particle-Filter RNN for object localization. In particular, it builds upon existing PFRNN work and incorporates additional assumptions from classical estimation methods.

Some of the reviewers find the presented approach theoretically sound and well explained.

However, two of the reviewers find the paper a minor modification of already published work of PFRNN. Some of the reviewers find the empirical results not strong enough, on not challenging setups. Further, one of the reviewer points out model size issues with the approach.

**Justification For Why Not Higher Score:**

Two reviewers recommend reject (1, 3) and one reviewer accept (8). We believe the accept reviewer isn't familiar with prior work, while the two reject reviewers point out the marginal novelty over published work as well as issues with the approach. Hence, unfortunately the paper is rejected from ICLR 2024.

**Justification For Why Not Lower Score:**

N/A

---

### Decision · Program_Chairs · 2024-01-16

Reject